# Retinal Manifestations in Patients with COVID-19: A Prospective Cohort Study

**DOI:** 10.3390/jcm11071828

**Published:** 2022-03-25

**Authors:** Eleonora Riotto, Vladimir Mégevand, Alexis Mégevand, Christophe Marti, Jerome Pugin, Alexandros N. Stangos, Leonardo Marconi Archinto, Gordana Sunaric Mégevand

**Affiliations:** 1Faculty of Medicine, University Hospital Geneva, 1205 Geneva, Switzerland; alexis.megevand@etu.unige.ch; 2Department of Surgery, University Hospital Geneva, 1205 Geneva, Switzerland; vladimir.megevand@gmail.com; 3Department of Internal Medicine, University Hospital Geneva, 1205 Geneva, Switzerland; christophe.marti@hcuge.ch; 4Department of Intensive Care, University Hospital Geneva, 1205 Geneva, Switzerland; jerome.pugin@hcuge.ch; 5Clinical Eye Research Centre A. Rothschild, 1205 Geneva, Switzerland; astangos@hotmail.com (A.N.S.); g.su.meg@gmail.com (G.S.M.); 6Department of Biology, Faculty of Science, Geneva University, 1205 Geneva, Switzerland; leonardo.marconi@etu.unige.ch

**Keywords:** retina, SARS-CoV-2, COVID, cotton wool spot, OCT, retinopathy, retinal hemorrhages

## Abstract

The recent outbreak of the Coronavirus SARS-CoV-2 has been declared a worldwide pandemic. Within various multi-organ involvement, several ocular manifestations have been described, such as conjunctivitis and retinopathy. The prevalence and severity of retinal lesions and their relation to the severity of the systemic disease are unknown. We performed a prospective, observational study on 172 consecutively hospitalized patients with acute confirmed COVID-19 infection. All patients underwent screening widefield fundus photography at the time of hospital admission. Despite no ocular or vision-related symptoms, we found cotton wool spots (CWS) and/or hemorrhages in 19/172 patients (11%). Diabetes history, overweight, and elevated C-reactive protein were more frequently observed among patients with retinal abnormalities, while a history of systemic hypertension was more frequently observed among patients without retinal findings. At a 3-month follow-up visit, CWS had subsided in all patients.

## 1. Introduction

In December 2019, a new disease, Severe Acute Respiratory Syndrome-Coronavirus-2 (SARS-CoV-2), emerged from China. The virus quickly led to a pandemic disease named coronavirus disease 2019 (COVID-19) [1]. Currently, there are about 140 million confirmed infections with SARS-CoV-2 worldwide and approximately 3,000,000 deaths associated with COVID-19 [2]. SARS-CoV-2 infection appears to involve multiple organ systems with pathological manifestations, including the lung, heart, kidneys, and brain [3]. It has been hypothesized that these organs, and their associated blood vessels, may suffer from injuries caused by direct viral infiltration of endothelial cells [4,5], localized cytokine release [6], and alterations of the micro-vascularization, including a pro-thrombotic state, which is associated with an increased mortality rate [7]. Olfactory nerve involvement and gustatory dysfunction have been reported in early case series, with a pooled estimated prevalence of, respectively, 26.4% and 27.2% [8].

Similarly, alterations of the retina in patients with SARS-CoV-2 have been described. The most frequent are cotton wool spots (CWS) reported in several cross-sectional studies [8,9,10,11,12]. Other manifestations include retinal hemorrhages, dilated veins, and tortuous vessels [13]. The pathophysiology, incidence, and clinical course of these retinal findings remain unclear.

The objective of this study was to perform a systematic screening for retinal alterations among consecutive patients hospitalized for an acute COVID-19 infection requiring supplemental oxygen at a tertiary university hospital and provide an estimate of retinal involvement prevalence and a description of the observed alterations. Furthermore, we sought to identify clinical features associated with the presence of retinal abnormalities and to study the longitudinal evolution of the identified lesions.

## 2. Materials and Methods

This was a longitudinal, prospective, observational study conducted at the University Hospitals of Geneva (HUG), Switzerland, in collaboration with the Clinical Eye Research Centre Adolph de Rothschild (CERC). We prospectively invited into the study all consecutive patients admitted to the Department of Internal Medicine and Medical Intermediate Care Unit of HUG between 15 March and 15 June 2021. Patient recruitment and screening were conducted at the Department of Internal Medicine and Medical Intermediate Care Unit of HUG. Upon discharge, follow-up was performed at CERC.

Patients were invited into the study according to the following inclusion criteria: adult patients (age >18 years) with acute SARS-CoV-2 infection (confirmed by a positive RT-PCR test), requiring oxygen supplementation (oxygen saturation of 89%), requiring an inspired fraction of oxygen superior to 50%. In accordance with institutional and international recommendations [11], all patients received continuous oxygen supplementation when required via nasal cannula or High-Flow Nasal Oxygen when indicated. Dexamethasone (6 mg daily) was recommended for patients with acute SARS-CoV-2 infection requiring supplemental oxygen.

This study followed the principles of the Declaration of Helsinki and was approved by the HUG Ethics Committee (ID 202002873). All participants signed the informed consent form prior to enrollment. Patients unwilling or incapable of signing the informed consent form and patients unable to sit to allow for fundus photography were excluded from the study.

### 2.1. Procedure

On first day of admission, all patients fulfilling the inclusion criteria underwent non-mydriatic widefield fundus photography (Optos, Daytona, Optos PLC, Dunfermline, UK) in both eyes. Detailed ophthalmological history was taken, in particular in regard to any potential vascular retinopathy such as diabetes, systemic hypertension, or others. Vital parameters and biological variables were collected on the day of admission. Concomitant diseases were listed.

All fundus photos were analyzed on the same day by two experienced ophthalmologists (AS and GSM). Patients with fundus photographs showing retinal anomalies of any kind were asked, according to the protocol, to undergo a second full eye examination, including fundus photographs and optical coherence tomography (OCT) (DRI OCT Triton, Topcon, Tokyo, Japan). Three-month ophthalmic follow-up was scheduled.

We performed a detailed evaluation of the presence of retinal lesions, their correlation to possible preexisting comorbidities, and the possible correlation with the severity of COVID-19 infections.

### 2.2. Outcomes

The primary aim of our study was to evaluate the early presence and prevalence of retinal lesions in a sample of COVID-19 infected patients suffering from a moderate disease admitted to the ward or intermediate care unit. The secondary aim of our study was to evaluate the possible correlation between retinal anomalies and patients’ clinical or biological characteristics.

### 2.3. Statistical Analysis

Patient characteristics were reported with means and standard deviations as appropriate. Between-group comparisons between patients with or without retinal abnormalities are provided using Mann–Whitney test for between-group values comparison and chi2 for proportions comparisons. The *P*-value for statistical comparison is 0.05 for all comparisons. Given the uncertainty regarding the prevalence of retinal abnormalities, uncertain evolution of the epidemic at the time of recruitment, and the limited availability of the fundus photographer, 2-month recruitment was planned without formal sample calculation.

## 3. Results

During the 2-month study period, a total of 172 patients were enrolled in the study. Eighty-nine were males, and eighty-three were females; the average age (±SD) was 55 years old (±11.7), and 17 patients reported diabetes history (9%), 52 reported systemic arterial hypertension (30%), 21 reported dyslipidemia (12%), and 11 reported coronary disease history (6%) (Table 1).

Ophthalmological screening was performed after a median of 1 day (range: 1 to 3 days) from hospitalization. All subjects received 6 mg of dexamethasone daily since their admission. On the day of fundus examination, none of the patients reported ocular symptoms.

### 3.1. Retinal Findings in COVID-19 Patients

Of the 172 examined patients, all of them had good-quality photos of both eyes. Retinal lesions were found in 19 patients (11%). Cotton wool spots alone (CWS) were found in 10 patients (6%), hemorrhages alone were found in 3 patients (2%), and CWS with hemorrhages were found in 6 patients (3.5%). Among these 19 patients, 6 had a bilateral presentation (3.5%). Typically, CWS were located at the posterior pole (Figure 1).

There were no signs of macular edema, optic nerve head, or peripheral retinal involvement. OCT scans showed hyperreflectivity of the inner retinal layers, consistent with CWS (Figure 1).

### 3.2. Evolution and Follow-Up

Three months after screening, there was a complete resolution of clinically evident CWS (Figure 2) with concomitant normalization of OCT scan (Figure 2) in all patients. All patients were symptom-free 3 months after screening.

### 3.3. Correlation between Retinal Findings and Clinical Features in COVID-19 Patients

Demographics, clinical features, and laboratory parameters of the 19 patients with observed retinal findings are reported in Table 2.

All variables were compared to those of subjects with no observed lesion. As shown in Table 1, diabetes history and obesity were significantly more prevalent among patients with retinal abnormalities, while hypertension was more frequently reported among patients without retinal lesions. Moreover, patients with retinal findings presented higher C-reactive protein (CRP) values (122 vs. 76 mg/L, *p* = 0.025) and potassium levels (4.2 vs. 3.9 mmol/L, *p* = 0.002).

## 4. Discussion

In the present study, CWS and/or hemorrhages were observed in 11% (19/172) of patients admitted to the hospital with acute COVID-19 infection in the absence of ocular or vision-related symptoms. Among the latter, ten presented only CWS (6%), three presented only hemorrhages (2%), and six presented both types of lesions (3.5%). Six patients (3.5%) had a bilateral presentation. Retinal lesions were located in the posterior pole in all cases without evidence of macular edema, optic nerve head, or peripheral retinal involvement. Retinal findings subsided completely 3 months after screening for COVID-19 retinopathy. All affected patients were symptom-free throughout the study period. We also observed that diabetes history and obesity were significantly more prevalent among patients with retinal abnormalities, while hypertension was more frequently reported among patients without retinal lesions.

CWS are non-specific, acute retinal lesions representing a focal ischemic process due to an occlusion of a terminal retinal arteriole. The exact physiopathology, incidence, and behavior of the CWS found in relation to COVID-19 is not yet well understood.

One explanation could be a direct viral tissue injury and infiltration of the endothelial cells causing vascular obstruction. In fact, it is reported that one of the major players for SARS-CoV-2 entry into the host cell is the angiotensin-converting enzyme 2 (ACE2) receptor, which is expressed in the lung, heart, kidney, intestine, and, most relevantly, the retina [4,14,15,16]. Additionally, the resulting cytokine release might cause further direct injury with the release of inflammatory and apoptosis-inducing mediators. Subsequent localized microvascular inflammation, a form of vasculitis or an immune-complex deposition on vessel walls, similar to findings in human immunodeficiency virus [12], could trigger endothelial activation, vasodilation, and pro-thrombotic conditions.

Another explanation could be the occlusion of retinal pre-capillary arterioles caused by a hypercoagulable state in relation to a disproportionate production and deposition of fibrin clots in small- and mid-sized vessels. Such findings have been observed in disseminated intravascular coagulopathy-like syndrome [17]. The significantly increased levels of D-dimer, fibrinogen, and CRP observed in our patients support this hypothesis. Moreover, the localization and features of CWS observed in our study are similar to those described in Purtscher-like retinopathy (PLR), a syndrome related to a hypercoagulability state related to various systemic diseases [18]. CWS in PLR are usually bilateral with or without retinal bleeds. Systemic steroids and hyperbaric oxygen have been shown to be beneficial in Purtscher-like retinopathies [19]. Thus, it could be possible that COVID-19-related retinopathy could be part of the PRL-like syndrome caused by a state of hypercoagulability. A recent study performed Optical Coherence Tomography Angiography (OCT-A) on patients fully recovered from COVID-19 suggested that thrombotic microangiopathy could explain persistent retinal vascular changes [20]. Another study reported an inversely proportional correlation between D-dimer levels and superficial retinal vascular density measured with OCT-angiography among COVID-19 patients [21]. The authors attributed these findings to a hypercoagulability state. Nevertheless, more than 70% of the patients included in their study presented comorbidities (such as diabetes, arterial hypertension, or dyslipidemia), which could have had an impact on the retinal vascular presentation.

One last hypothesis could be that CWS manifest following diffuse or focal vasospasm caused by hemodynamic disturbances. Hypoxia related to acute respiratory distress or to an acute rise in blood pressure in the early phase of the disease could be the cause of such a condition. It is yet undetermined if vascular changes following the acute respiratory failure in COVID-19 may mimic a chronic condition such as sleep apnea. Sleep apnea is often associated with systemic hypertension causing retinal arteriolar vasoconstriction. Thus, focal CWS could be explained by focal anoxia in some susceptible terminal arterioles. Nevertheless, vasoconstriction related to sleep apnea manifests as part of a chronic condition. A recent study supports our results by demonstrating a significant reduction in vessel density of the superficial and deep capillary plexus in patients who had recovered from COVID-19 compared to those in healthy subjects [19,22,23].

To our knowledge, this is the first study that performed fundus examination in patients with COVID-19 and subsequently conducted a long-term follow-up of patients with visualized lesions. This study also highlighted a statistically significant correlation between CWS lesions and diabetes history and obesity. Considering that retinal vasculature can be directly evaluated via fundus examination, visualization and quantification of such lesions could represent signs of vascular involvement in other organs besides the eye and, therefore, a new biomarker related to COVID-19. Thus, the above-mentioned comorbidities could be a severe risk factor not only for the retinal vasculature but also for the brain and other organs’ vasculature.

The main strength of this study was that we performed a prospective, observational study during which we screened patients at an early stage of their symptomatic disease. Moreover, we performed follow-up investigations on patients with detected lesions in order to characterize the evolution and duration of such lesions.

On the other hand, the limitation of this study was that, unfortunately, because of the complexity of executing a fundus examination on intubated patients, we were not able to investigate the vascular impact of SARS-CoV-2 in severely ill patients in the ICU.

## Figures and Tables

**Figure 1 jcm-11-01828-f001:**
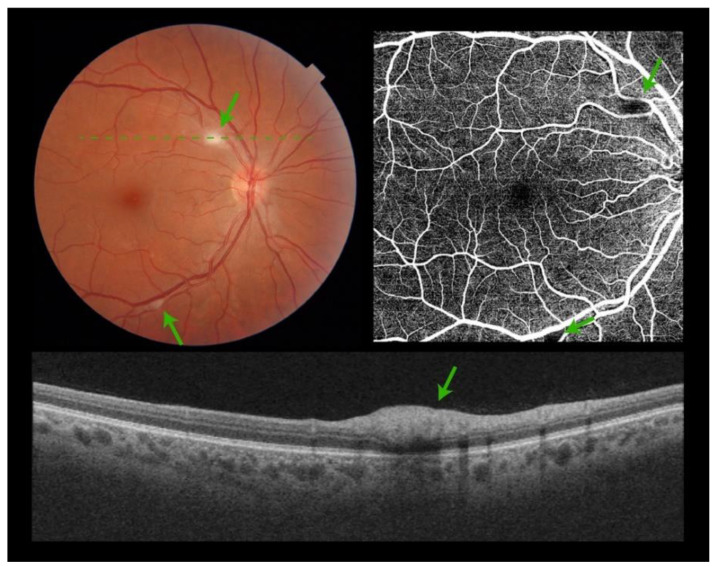
Composite figure. Color picture showing the right eye retina of a patient admitted to hospital with acute SARS-CoV-2 infection. Two isolated cotton wool spots are visible at the posterior pole (**green arrows**). No other signs of retinopathy can be detected (**top left**). Optical Coherence Tomography Angiography showing a black shadow (**green arrows**) at the areas of the cotton wool spots. No other microvascular pathology can be identified (**top right**). Optical Coherence Tomography scan over the cotton wool spot (**green dashed line**) showing hyperreflectivity of the inner retinal layers (**bottom**).

**Figure 2 jcm-11-01828-f002:**
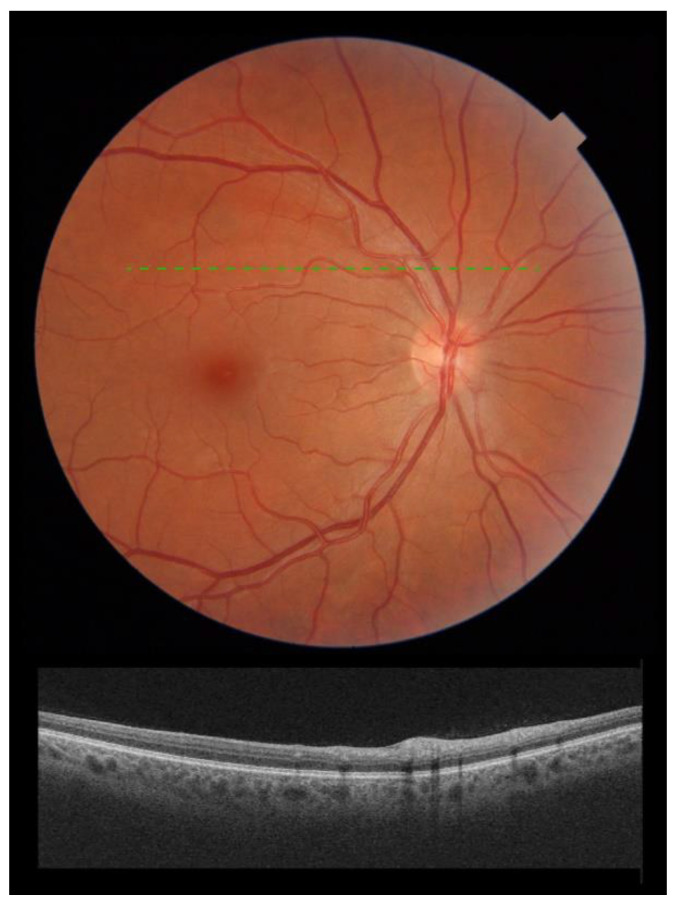
Composite figure. Color picture showing the right eye retina of a patient 3 months following admission to hospital with acute SARS-CoV-2 infection. The cotton wool spots detected at admission had subsided. No other signs of retinopathy could be detected (**top**). Optical Coherence Tomography scan over the area previously occupied by the cotton wool spot (**green dashed line**) showing normalization of the inner retinal architecture (**bottom**).

**Table 1 jcm-11-01828-t001:** Demographics, clinical features, and laboratory parameters of the 153 patients without retinal findings and of the 19 patients with observed retinal findings.

	Patients with No Retinal Anomalies N = 153	Patients with Observed Retinal Anomalies N = 19	*p*-Value
Age, mean years (SD, range)	55.1 (12.1, 43–67.2)	54.7 (11, 43.7–65.7)	0.890
Male gender, n (%)	76 (49.7%)	13 (68.4%)	0.197
Hypertension, n (%)	50 (32.70%)	2 (10.50%)	**0.048**
Smoking, n (%)	19 (12.40%)	4 (21%)	0.300
Coronary disease n (%)	10 (6.50%)	1 (5.30%)	0.840
Dyslipidemia, n (%)	19 (12.40%)	2 (10.50%)	0.812
Diabetes, n (%)	10 (6.50%)	7 (36.80%)	**<0.0001**
Overweight ^a^, n (%)	33 (21.60%)	8 (42.10%)	**0.049**
Pulse rate, mean (SD)	89.4 (23.2)	96.4 (10.49)	0.377
Systolic blood pressure (mmHg), Mean, (SD)	126 (23.2)	130 (20.4)	0.6477
Temperature (°C), Mean, (SD)	37.8 (0.92)	37.7 (0.82)	0.752
Oxygen saturation, Mean, (SD)	93.7 (3.88)	94.6 (4.02)	0.382
Hemoglobin (g/L), mean (SD)	137.3 (19.2)	139.5 (20.32)	0.649
White blood cells (g/L), mean (SD)	6.9 (3.03)	6.8 (2.28)	0.870
Thrombocytes (g/L), mean (SD)	227.9 (93.7)	239.4 (101.62)	0.629
Fibrinogen (g/L), mean (SD)	5.4 (1.51)	5.1 (0.87)	0.527
CRP (mg/L), mean (SD)	76.3 (72.26)	121.9 (122.35)	**0.025**
Creatinine (µmol/L), mean (SD)	72.3 (22.19)	79.7 (20.59)	0.179
Bilirubin (µmol/L), mean (SD)	7.9 (3.92)	7.1 (2.96)	0.551
D-dimer (µmol/L), mean (SD)	1073.3 (1023.1)	1185.7 (1024.69)	0.678
PH, mean (SD)	7.46 (0.04)	7.42 (0.15)	**0.042**
pCO2 (kPa), mean (SD)	4.8 (0.89)	4.9 (1.10)	0.840
pO2 (kPa) mean (SD)	8.2 (3.15)	7.0 (4.01)	0.330
Oxyhemoglobin (%), mean (SD)	79.3 (24.89)	74.2 (23.68)	0.584
Lactate (mmol/L), mean (SD)	1.3 (0.68)	1.6 (0.58)	0.367
Bicarbonate (mmol/L) mean (SD)	25.1 (3.72)	22.7 (7.65)	0.135
Glycaemia (mmol/L), mean (SD)	8.2 (3.57)	10.6 (5.80)	0.152

^a^ Overweight defined as a body mass index (BMI) (kg/m^2^) over 25 kg/m^2^.

**Table 2 jcm-11-01828-t002:** Demographics, clinical features, and laboratory parameters of the 19 patients with observed retinal findings.

Sex	Age	Comorbidities	Type of Lesions at Baseline	Eye Right/Left/	Type of Lesions at 1st Follow-up	Type of Lesions at 2nd Follow-Up	Type of Lesions at 3rd Follow-Up	Oxygen Saturation (%)	Red Blood Cells (G/L)	Hemoglobin (g/L)	White Blood Cells (G/L)	Thrombocytes (G/L)	CRP (mg/L)	Sodium (mEq/L)	Potassium (mEq/L)	Creatinine (µmol/L)	D-Dimer (µmol/L)
			(Day 1)		(Day 38)	(Day 72)	(Day 104)										
F	72	Obesity, diabetes	CWS	R, L	None	None	n/a	90	4.1	125	4.7	223	13	136	4.6	73	1458
M	46	None declared	CWS	L	CWS	None	n/a	93	5	148	8.1	220	114	137	5.2	61	675
F	64	None declared	CWS/bleeds	R/L	None	n/a	n/a	97	3.96	104	6.4	423	12.2	137	4	54	4357
F	58	Obesity, diabetes, hypertension	CWS	L	None	None	n/a	82	5.8	127	6.8	229	140	132	4.4	139	2379
M	65	Hypercholesterolemia	CWS, bleeds	L	CWS	None	n/a	96									506
F	63	Diabetes	CWS/bleeds	R/L	CWS	None	None	100	5.66	159	7.8	192	44	133	5.3	107	688
M	43	None declared	CWS	L	None	None	n/a	95	5.85	173	11.8	339	250	134	4.5	86	347
F	68	Hypertension, hypercholesterolemia, obesity, diabetes	CWS	R, L	CWS	None	None	96	4.7	134	8.1	271	59.2	131	3.9	71	1088
F	42	None declared	CWS	R	None	None	n/a	94	4.6	133	3.8	179	75.8	141	3.8	70	1174
M	51	Diabetes	CWS/hard exudates, bleeds	R/L	CWS, hard exudates, bleeds	Hard exsudates, bleeds	None	98	5.33	158	7.3	230	60.8	131	4.3	84	439
H	51	Obesity	CWS, bleeds	L	None	None	None	92	4.85	147	5.5	99	180	137	4.3	75	1235
H	36	Obesity	CWS	L	None	n/a	n/a	95	5	141	3.3	344	62	137	3.5	73	n/a
H	46	Obesity	CWS	R	None	None	n/a	96	5.6	165	9	457	33	139	4.9	82	1908
M	57	Diabetes	CWS	R	None	n/a	n/a	94	5.02	144	9.3	296	280.86	134	4.3	70	1202
M	71	Obesity, diabetes	CWS/bleeds	R/L	None	n/a	n/a	100	2.63	89	6.6	106	116.37	130	4.4	66	n/a
M	51	Obesity	Bleeds	R, L	None	n/a	n/a	94	4.78	148	5.4	138	491.58	138	3.7	59	>10,000
M	48	Obesity	Bleeds	R/L	None	n/a	n/a	92	4.77	142	7.7	158	89.75	133	3.7	92	581
M	62	Obesity	CWS	L	None	n/a	n/a	98	4.4	141	7.5	256	n/a	134	4	104	590
F	45	Obesity	Bleeds	L	None	n/a	n/a	95	5.19	133	2.7	150	50.61	136	3.2	68	344

## Data Availability

All data generated or analyzed during this study are included in this article. Further inquiries can be directed to the corresponding author.

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
