# Peer review of "Retinal Manifestations in Patients with COVID-19: A Prospective Cohort Study"

_jcm, 2022, doi:10.3390/jcm11071828_

Round 1
Reviewer 1 Report
The authors cover an important ophthalmic aspect of retinal manifestation in moderately severe COVID -19 infection. The manuscript is well written.
There are a few comments
Page 1 line 37 : “Olfactory nerve” instead of “Olfactive nerve”
Page 2 line 69 : “incapable” instead of “uncapable”; line 70 “unable to sit” instead of “unable of sitting”
Line 77 “Concomitant disease”….change to “diseases”
Line 88: “…aim of our study is to evaluate…”- “was” instead of “is”
Line 91: “…patients…”- change to patient’s or patients’
Line 94: “Patient’s characteristics are reported…”- use “were”
Line 105: remove period before “17”
Line 161: “CWS are non- specific, acute retinal lesions … occlusion of a terminal retinal arteriole.” – lines are discontinuous, kindly correct it.
Figures 1 and 2 can be improved by merging the images uniformly and drawing a line on the fundus image where the line scan has been taken
A conclusion statement can be added.
Author Response
Thank you for your kind answer!
We corrected all the listed points.
Reviewer 2 Report
Riotto et al. conducted a clinical study of potential correlations of hospitalized COVID–19 patients with retinal lesions. They aimed to provide an estimate of retinal involvement prevalence, and to identify clinical features associated with the presence of retinal abnormalities. To do so, the authors managed to recruit 172 patients over a 2 months study period, and performed screening fundus wide field photography at the time of hospital admission, and followed by a 3-month follow-up. Overall, this study was well conducted, of great interest to the field, and need minor writing check.
Author Response
Thank you very much !
Reviewer 3 Report
The manuscript entitled “Retinal manifestations in patients with COVID-19: A prospective cohort study” deals with a longitudinal, prospective, observational study, screening for retinal alterations among consecutive patients hospitalized for an acute COVID-19 infection requiring supplemental oxygen. The study was performed on a total of 172 patients
The manuscript is well written and easy to understand. The experiments carried out were enough and suitable for the purpose of the manuscript. The references used in the manuscript are recent and adequate. Regarding the novelty of the manuscript, as far as I am concerned this is the first time that the fundus examination in patients with COVID-19 and subsequently conducted a long-term follow-up of patients with visualized lesions.
In my opinion, the results shown in the present manuscript are interesting for a broader community, nonetheless. Despite its great potential, the authors should revise some small issues to improve the readability of the manuscript.
- Keywords are missing.
- Figure 1 is confusing, it refers to A and B, while there are three images.
- Table 1 is confusing, try to improve it, the use of comma between the parameter and the unit shown is not consistent.
- Table 1 and 2 lack title, Follow the instructions for authors “Tables should have a short explanatory title and caption”.
- Line 162: I guess there is a typo. ¨
- Revise the references section, there are several references attached to [3]; and other references are not in format [7], [13]; Follow the instructions for authors in the references section.
Best regards
Author Response
Thank you.
We changed everything as asked.